# Non-Invasive Detection of a De Novo Frameshift Variant of *STAG2* in a Female Fetus: Escape Genes Influence the Manifestation of X-Linked Diseases in Females

**DOI:** 10.3390/jcm11144182

**Published:** 2022-07-19

**Authors:** Aldesia Provenzano, Andrea La Barbera, Francesco Lai, Andrea Perra, Antonio Farina, Ettore Cariati, Orsetta Zuffardi, Sabrina Giglio

**Affiliations:** 1Medical Genetics Unit, Department of Experimental and Clinical Biomedical Sciences “Mario Serio”, University of Florence, 50139 Florence, Italy; 2Unit of Medical Genetics, IRCCS Ospedale Policlinico San Martino, 16132 Genoa, Italy; andrea.labarbera@hsanmartino.it; 3Medical Genetics Unit, Department of Medical Sciences and Public Health, University of Cagliari, 09124 Cagliari, Italy; francescolai.md@gmail.com (F.L.); sabrinar.giglio@unica.it (S.G.); 4Unit of Oncology and Molecular Pathology, Department of Biomedical Sciences, University of Cagliari, 09124 Cagliari, Italy; andrea.perra@unica.it; 5Division of Obstetrics and Prenatal Medicine, Department of Medicine and Surgery (DIMEC), IRCCS Azienda Ospedaliero-Universitaria di Bologna, 40126 Bologna, Italy; antonio.farina@unibo.it; 6Villa Cherubini Hospital, 50133 Florence, Italy; ettore.cariati@gmail.com; 7Department of Molecular Medicine, University of Pavia, 27100 Pavia, Italy; orsetta.zuffardi@unipv.it; 8CeSAR, University Service for Research, University of Cagliari, 09124 Cagliari, Italy

**Keywords:** X-linked diseases, non-invasive whole exome sequencing, fetal cell-free DNA, escapee genes, X-inactivation

## Abstract

Background: We report on a 20-week-old female fetus with a diaphragmatic hernia and other malformations, all of which appeared after the first-trimester ultrasound. Methods and Results: Whole trio exome sequencing (WES) on cell-free fetal DNA (cff-DNA) revealed a de novo frameshift variant of the X-linked *STAG2* gene. Loss-of-function (LoF) *STAG2* variants cause either holoprosencephaly (HPE) or Mullegama–Klein–Martinez syndrome (MKMS), are de novo, and only affect females, indicating male lethality. In contrast, missense mutations associate with milder forms of MKMS and follow the classic X-linked recessive inheritance transmitted from healthy mothers to male offspring. *STAG2* has been reported to escape X-inactivation, suggesting that disease onset in LoF females is dependent on inadequate dosing for at least some of the transcripts, as is the case with a part of the autosomal dominant diseases. Missense *STAG2* variants produce a quantity of transcripts, which, while resulting in a different protein, leads to disease only in hemizygous males. Similar inheritance patterns are described for other escapee genes. Conclusions: This study confirms the advantage of WES on cff-DNA and emphasizes the role of the type of the variant in X-linked disorders.

## 1. Introduction

Prenatal whole exome sequencing (WES) in the evaluation of fetuses with congenital anomalies allowed genetic diagnoses in at least 10% of cases in which the standard chromosomal tests were negative [1,2]. To date, WES analyses are mainly carried out on DNA obtained from chorionic villi or amniocytes; however, the increasing reliability of molecular investigations on cell-free DNA from blood sampling of pregnant women makes the non-invasive approach more attractive (despite the difficulties in distinguishing maternal variants from the fetal ones) [3]. It has been demonstrated that trio-whole exome sequencing (trio-WES) using fetal cell-free DNA (cff-DNA) can be analyzed with sufficient sensitivity, at least in the presence of a positive family history of malformations [4,5].

In this report, we show that trio-WES on cff-DNA is a useful approach for detecting the molecular cause of fetal congenital malformations, even in couples with no apparent risk of genetic disease, and that this procedure is further recommendable in light of the superior quantity and quality of the DNA extracted from plasma compared to that of amniocytes. With this approach, we identified a de novo frameshift variant in *STAG2*, an X-linked gene already associated with congenital malformations, some of which are compatible with the ultrasound abnormalities of the female fetus we studied. As already shown [6], *STAG2* missense mutations affect only males in which the variant is de novo or inherited from healthy mothers, while LoF mutations result in only affected heterozygous females, in which the variant is de novo.

This type of pattern is common to various X-linked genes that escape inactivation, such as *DDX3* [7], *BCOR* [8], *KDM6A* [9], *OFD1* [10], as is the case for *STAG2* [11]. Therefore, the presence of malformations in females with *STAG2* LoF variants, as well as in other cases of LoF variants in X-linked escapee genes, may not be related to the skewing of inactivation but rather depend on the dosage of residual transcripts expressed by both Xs, the wild type and the mutated. Conversely, in males, the amount of transcripts expressed by the single mutated X is below the embryo survival threshold.

We emphasize how highlighting the variant type in escapee genes is the key to proper genetic counseling for prospective parents.

## 2. Materials and Methods

After genetic counselling, trio-WES analysis on fetal DNA extracted from amniotic fluid (AF) and on cff-DNA was performed with a high-resolution SNP-CGH array. Both parents, who are of Italian ethnicity, signed informed written consent before blood collection.

### 2.1. Genomic DNA (gDNA) Extraction

Fetal DNA was obtained from amniotic fluid, and cff-DNA was extracted from maternal plasma. Parental blood DNA was extracted from peripheral blood using a QIAamp DNA Mini Kit (Qiagen, Hilden, Germany).

### 2.2. SNP-CGH Array

A total of 200 ng of DNA from the amniotic fluid and the parents’ peripheral blood was hybridized to the Infinium_CytoSNP_850K genotyping array (Illumina, Inc., San Diego, CA, USA). Genome coordinates were based on Human Genome, Feb 2009 Assembly (GRCh37, hg19). The parental origins of all chromosomes were determined by counting informative SNPs on an Excel file, containing LogR and B allele frequency (BAF) values for each array probe, produced by Genome Studio Software. BAF of 0 signifies the genotype (A/A or A/–), where 0.5 signifies (A/B) and 1 signifies (B/B or B/–). Different BAF values were displayed for AAB and ABB genotypes.

### 2.3. Cff-DNA Extraction

A total of 10 mL of peripheral venous blood was collected from the pregnant woman in a Cell-Free DNA Collection Tube (Roche). Blood samples were first centrifuged at 1160 g for 10 min at 4 °C to separate the plasma from peripheral blood cells. The plasma portion was carefully transferred into 3 tubes (1.5 mL) and subjected to centrifugation at 18,620 g for 2 min at 4 °C to pellet the remaining cells. Cell-free DNA from 4.5 mL of maternal plasma was extracted using the QIAamp Circulating Nucleic Acid Kit (Qiagen) following the protocol.

### 2.4. Library Construction

DNA enrichment was performed after end-repairing and before adaptor ligation during cff-DNA library construction while gDNA was previously fragmented by enzymatic reaction (KAPA HyperPlus Kits-KAPA Biosystem, Roche, Basel, Switzerland). Magnetic beads were used for the purpose of size-selecting the end-repaired DNA fragments with sizes smaller than 250 bp. The supernatant containing size-selected DNA fragments was then transferred to another tube for adaptor ligation.

### 2.5. Sequencing

DNA library concentration was determined by a Quantus fluorometer (Promega) measurement. For DNA sequencing, the cff-DNA library was captured with the NimbleGen SeqCap EZ Exome v3 capture kits (Roche) as well as genomic DNA, and sequenced using NextSeq 550 Illumina System with paired-end sequencing mode producing raw sequencing reads with sizes up to 300 bp.

### 2.6. Estimation of Cff-DNA Fraction

To determine the fetal DNA fraction in the maternal plasma, since the fetus was a female and, therefore, devoid of the Y chromosome, we analyzed all high-frequency variants of paternal origin that were absent in the maternal DNA, and, in particular, the X chromosome variants exclusively of paternal origin.

### 2.7. Investigation for the Presence of a Mosaic in the Parents

To assess the possibility of mosaicism of the causative variant identified by WES and *STAG2* c. 3458_3459 delTT (p.Leu1153ArgfsTer3) in the blood of one of the parent’s, we performed high-depth amplicon sequencing. We used specific primers (forward-TACGGCCTGAGGATAGCTTC, reverse-CATGTTACCTGCGTGCTTCA) to amplify a 240 bp region in which the identified variant was located. We ligated universal adapters to the PCR product; after purification, the amplification product was sequenced on NextSeq550.

### 2.8. Data Analysis

High-depth sequencing was performed to detect variants with low percentages, which is the same approach used when a mosaicism asset is suspicious. Variant calling was performed using Varscan software [12,13] to call variants that met desired thresholds for reading depth, base quality, variant allele frequency, and statistical significance. This software allows analyses of variants present in regions with low coverage, somatic mutations, and multi-sample variants, as well as germlines. All variants obtained were filtered in agreement with ultrasound malformations only.

## 3. Results

A fetal ultrasound (USA) in a 35-year-old Italian woman revealed multiple malformations despite first-trimester screening showing regular parameters (nuchal translucency, NT, 1.9 mm; crown-rump length, CRL, 7 mm). Additionally, the cell-free DNA screening test for fetal aneuploidy had a Z score in the normal range. The pregnancy was uneventful until the second level US examination at the 19th gestation week when the left-sided diaphragmatic hernia (CDH), dextroposition of the heart, and severe aortic coarctation associated with a ventricular septal defect, and a Dandy–Walker malformation (Figure 1 and Figure 2) were highlighted.

The SNP-array revealed a 46, XX constitution with 12q21.1 deletion (74307540_74417793, hg19) containing a portion of long intergenic non-coding RNAs (LINC02882), inherited from the healthy mother and reported on the Genomic Variants Database (DGV) with a frequency of less than 1% (3/17421). Given the severity of the fetal condition, the couple decided to terminate the pregnancy at 20 weeks. An autopsy showed a left diaphragmatic hernia with right hemidiaphragm, dextroposition of the heart, and the Dandy–Walker anomaly. Low-set ears were evident. The karyotype on the fetal tissue was 46, XX.

The cff-DNA exome data were analyzed together with those of the parents. The fetal fraction was calculated as 15–18% in accordance with the mean fetal fraction obtained in the laboratory for samples referring to 19 weeks of pregnancy (30 samples). cff-DNA reading was performed at a greater reading depth than the genomic one (cff-DNA coverage average 167X, gDNA coverage average 116X, Table 1).

Since the family history was negative and the ultrasound features of the fetus (diaphragmatic hernia, aortic coarctation, Dandy–Walker malformation) suggested a rare condition or neurodevelopmental disorder associated with these characteristics, we excluded both variants with a frequency greater than 1% and rare variants not associated with the phenotype.

WES identified a de novo heterozygous frameshift variant c.3458_3459delTT (p.Leu1153ArgfsTer3) located in exon 31 of the *STAG2* gene (NM_001042749; ID DECIPHER: Patient 482574) defined as pathogenic according to the American College of Medical Genetics (ACMG) guidelines [14]. This variant is not reported in GnomAD, ExAC, or dbSNP NFE (European non-Finnish) databases. Its pathogenicity is presumed because it is of the frameshift type and leads to a premature stop codon. Moreover, a bioinformatic tool (http://autopvs1.genetics.bgi.com/05/2022) predicted mRNA decay with a very strong probability [15]. The *STAG2* two-nucleotides deletion falls 9 base pairs upstream of the donor splice site and does not predict alteration of the splicing of exons 31 and 32, according to the BDGP prediction tool (https://www.fruitfly.org/index.html/05/2022) (Figure 3). It is therefore foreseeable that some transcripts provide a truncated protein predicted to undergo mRNA decay. The Leu1153ArgfsTer3 variant is found in a highly conserved region, presumably maintained by natural selection (Vertebrate Multiz Alignment and Conservation (100 Species), UCSC https://genome.ucsc.edu/05/2022).

The *STAG2* variant was only detected in approximately 9% of the reads, as expected for heterozygous variants with fetal DNA representing 15% of the total DNA. The mutation appeared to be de novo according to the parents’ genotype (Figure 4). The comparison of maternal and paternal informative polymorphisms related to *STAG2* and other X-linked genes strongly suggested that the variant allele was of maternal origin.

To exclude any cryptic mosaic for the variant in one of the parents, we performed a high-depth amplicon sequencing (759.350X) where the *STAG2* variant was not detected.

## 4. Discussion

In this study, we showed the feasibility of the DNA diagnosis in a 20-week female fetus with multiple congenital anomalies, including diaphragmatic hernia and cardiac abnormalities, all detectable starting from the second trimester. The identification of an unreported de novo frameshift variant in the X-linked *STAG2* gene matched well the clinical features of the fetus, some of which overlapped those reported in the MKMS patient. *STAG2* is a member of the cohesin complex, which is involved in numerous cellular processes, such as the separation of sister chromatids during cell division, gene expression, heterochromatin formation, and DNA repair [16,17].

Somatic LoF mutations of *STAG2* have been frequently observed in several types of human cancers [18]; more recently, germline variants have been associated with MKMS and HPE. MKMS (OMIM#301022) presents with highly variable clinical features, some of them common to other cohesinopathies, in particular Cornelia de Lange syndrome [19]. A diaphragmatic hernia, reported in some Cornelia de Lange syndromes (1 and 4), is the most evident malformation in the second trimester of prenatal life [20]. Cleft lips and palate are also common, together with global developmental delay, intellectual disability, and hypotonia. Hypoplastic left heart, possibly secondary to diaphragmatic hernia, was also frequently reported together with brain malformations and microcephaly, the latter usually only detectable in the third trimester of pregnancy. CDH was only reported in females with *STAG2* variants but not in males [6].

Genetic testing in prenatal CDH is highly recommendable for accurate counseling and evaluation of fetus eligibility to pre- and/or postnatal therapy. A genetic etiology for CDH, either pre- or postnatal, was found in about 10% of the cases by conventional/karyotype or CGH/SNP array [21] whereas WES in 39 trios showed predicted damaging de novo variants in 21% of complex cases and 12% of isolated ones [22,23]. Variants in genes belonging to the cohesion complex were consistently associated with CDH [24].

The pathogenetic mechanisms underlying cohesinopathies are still unclear but seem to involve different functions of the cell cycle, including the cohesion of sister chromatids [25]. STAG2, binding to RAD21, acts as a shield to WAPL, protecting the cohesion of sister chromatids during mitosis [26]. As for the second phenotype associated with variants of *STAG2*, i.e., HPEs, these are severe forms of lobar or semilobar holoprosencephaly with agenesis or dysgenesis of the corpus callosum [27]. HPE is also present in some patients with Cornelia de Lange syndrome associated with variants in *SMC3*, *RAD21*, and *SCM1A*. Mouse studies have shown that Stag1 and Scm1a are expressed in prosencephalic neural folds, consistent with forebrain morphogenesis and holoprosencephaly pathogenesis. The knockdown of these two genes causes aberrant expression of several HPE-associated genes [27].

So far, more than 20 cases with different pathogenetic variants of *STAG2* have been described both in hemizygous males and heterozygous females. In the four males, variants were of the missense type, de novo in two of the cases, and inherited from healthy mothers in the remaining two. In 14 females, LoF mutations were reported, with a 15th case having a missense variant. All female variants but one (case 4 in Kruszka et al., 27) were de novo. This has led to the conclusion that LoF variants are lethal in hemizygous males while causing multisystem congenital anomalies in females [6]. The X-inactivation in the two cases studied so far was skewed with preferential inactivation of the mutated X in the case with the most severe phenotype and of the wild type X in the case with less severe signs. Therefore, every possible relationship between X-inactivation and phenotype remains conflicting [6]. Interestingly, LoF variants in another X-linked gene involved in the cohesion pathway, *SMC1A*, which is another escape gene, act as those in *STAG2*. Both *STAG2* and *SMC1A* are reported as escapee genes [11,28]. By contrast, missense and LoF variants in the other X-linked gene involved in Cornelia de Lange syndrome, *HDAC8*, resulted in severely affected males with either healthy or mildly affected carrier mothers.

Mildly- or borderline-affected females with extreme skewing of X inactivation resulting in the silencing of the mutated *HDAC8* have also been reported [29], indicating that this gene does not escape X inactivation.

Studies on single-cell RNA sequencing [28] have shown that the escape from X-inactivation is often partial or incomplete and varies from tissue to tissue [7,28]. An example is provided by the genes in PAR1, which are expressed more strongly in males than in females, indicating that the total expression of the X chromosomes in females (active plus inactive X: Xa plus Xi) does not reach that deriving from the X and Y chromosomes in males where X inactivation does not occur [28]. Similarly, many of the X-Y homologous genes in non-PAR regions have increased expression. These data indicate that escapee genes do not totally escape chromosome inactivation [28].

Maintaining only a proportion of the activity of the active female X and male X, the final expression is mediated by X inactivation, the type of mutation, and the degree of expression in that specific tissue from either the Y or the active X. Therefore, it is likely that at least some of the diseases manifesting as semidominant are associated with LoF variants in escape genes, and as a consequence, the amount of protein product resulting from the combination of the expression of Xa and Xi allows the survival of the females but not of the hemizygous males [30]. In agreement, we expect that expression is related to escape genes rather than skewed inactivation, as shown for *DDX3X* and intellectual developmental disorders [7], *BCOR* and microphthalmia syndrome [8], *KDM6A* and Kabuki syndrome [9], *OFD1* and orofacial digital syndrome [10], and *USP9X* and neurodevelopmental disorders [31]. Indeed, all these genes are intolerant to LoF (gnomAD v2.1.1) and the lack of males with LoF variants, either resulting in total or partial loss of some of the transcripts, indicates that the mutation in the single allele is incompatible with embryo development as confirmed in Stag2- and Usp9x-null mouse embryos showing respectively mid-gestation and early embryo lethality [31,32].

The fact that many missense variants in some of the escapee genes instead result in diseases with X-linked recessive inheritance and, therefore, in the manifestation of the disease in males only, suggests that in these cases the altered gene retains some residual functions and produce sufficient amounts of crucial transcripts to survive embryonic development. In females, on the other hand, the quantity of transcripts from the wild type allele would be sufficient not to develop any or severe phenotype alterations [32] preventing the altered protein product from negatively interfering on that cellular pathway, as it happens for apparently healthy carriers of autosomal recessive diseases [33].

We also excluded that the apparently de novo variant we detected in the fetus hid parental somatic mosaicism at least in the blood by performing deep sequencing of the exon 31 amplicon where the *STAG2* variant was located. Some studies have in fact demonstrated the opportunity to perform these investigations for apparently de novo variants identified after exome sequencing in order to alert parents about the risk of recurrence of the disease in subsequent pregnancies [34,35].

In this study, we provided another example of the clinical use of the cff-DNA test to diagnose a de novo variant that causes a disease in a fetus. Furthermore, the minimum time for WES trio analysis (cff-DNA and parental blood) and confirmation of de novo variants (amplicon sequencing or digital PCR on cell-free DNA or variant sequencing on amniotic fluid) is less than three weeks, accomplishing the timing required by prenatal diagnosis.

Finally, we add further evidence to the concept that the inheritance model of X-linked recessive diseases does not fit with the female enrichment reported for some type of variants [28]. Indeed, dosage alteration in X-inactivation escape genes appears (to us) to be the best explanation for the paradox.

## Figures and Tables

**Figure 1 jcm-11-04182-f001:**
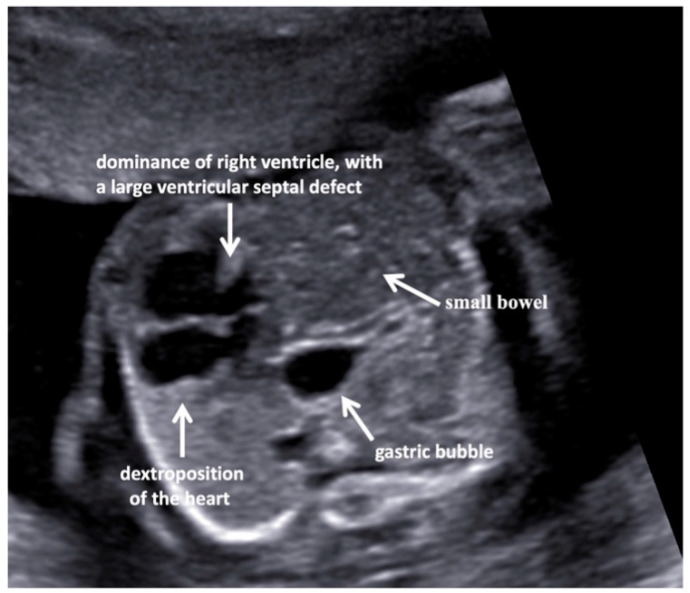
Axial view of the fetal thorax with dextroposition of the heart secondary to a diaphragmatic hernia on the left, at 20 weeks of pregnancy. The four-chamber view shows a dominance of the right ventricle, with a large ventricular septal defect. The gastric bubble is just behind the heart. The small bowel is visible within the left chest.

**Figure 2 jcm-11-04182-f002:**
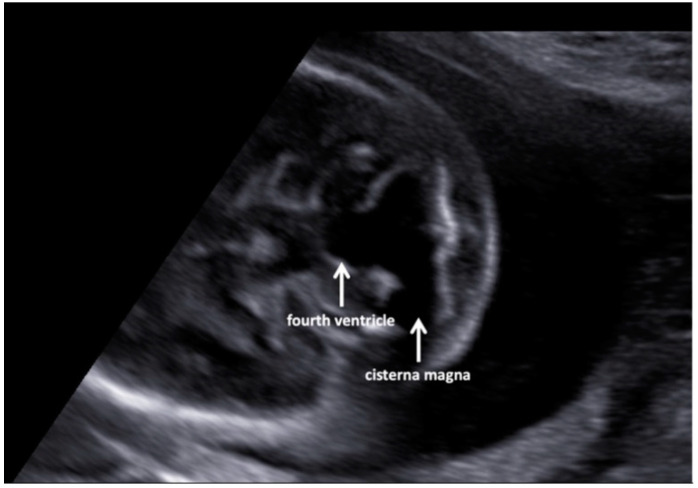
Axial views of the fetal head at the level of the mid-fourth ventricle (20 weeks of pregnancy), showing continuity of the fourth ventricle and cisterna magna.

**Figure 3 jcm-11-04182-f003:**
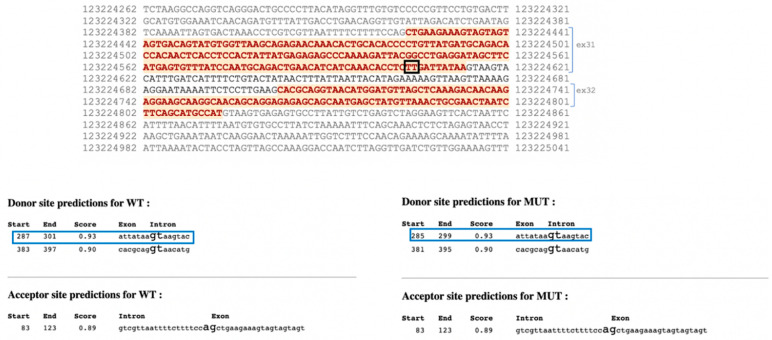
Top: Exon-intron sequences 31 and 32 of the *STAG2* gene. The coding regions are indicated in bold red. The two deleted bases, 9 nucleotides upstream of the splice donor site, are highlighted in the black box. Prediction of exon 31 splice site; the blue boxes indicate the canonical splice donor site that does not change in the mutated sequence.

**Figure 4 jcm-11-04182-f004:**
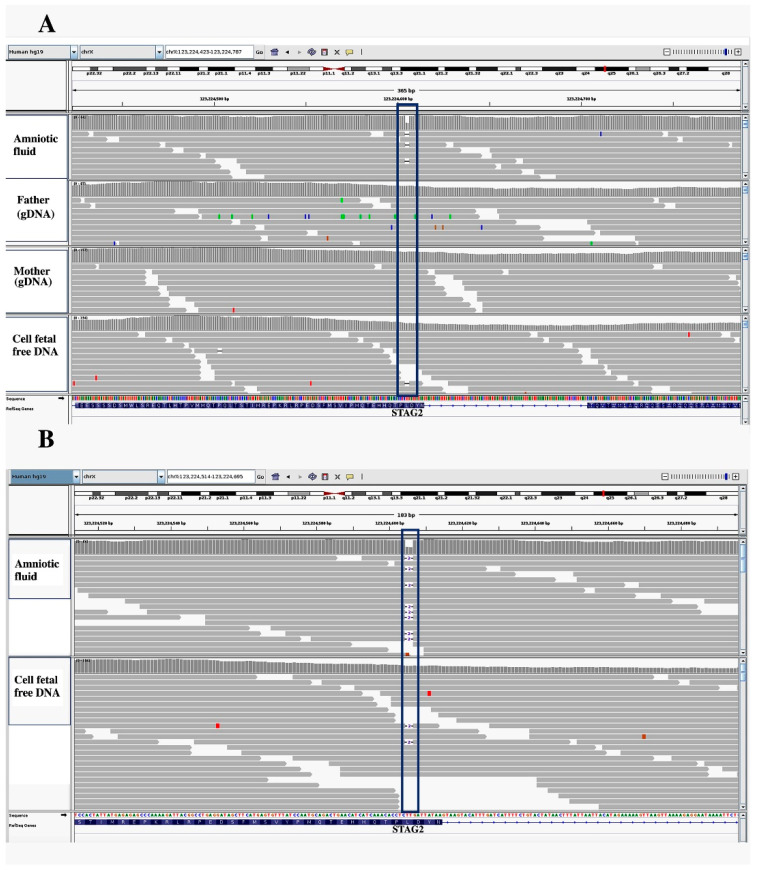
IGV representation of *STAG2* sequencing in cff-DNA and gDNA from amniotic fluid, father, and mother. the gray lines show the reads produced and aligned with the reference genome (hg19). (**A**) the *STAG2* heterozygous frameshift variant c. 3458_3459delTT was present in the cff-DNA and in the amniotic fluid but not in the parents. The blue boxes highlight the amniotic fluid and cff-DNA, the horizontal black lines represent the two base deletions as clearly indicated in the enlargement below. (**B**) Zoom of the *STAG2* variant in the amniotic fluid and cff-DNA; in amniotic fluid, the variant is more represented than in cff-DNA in which the dilution of the variant by maternal DNA makes its recognition less immediate.

**Table 1 jcm-11-04182-t001:** Depth and coverage of the target regions.

Sample	Coverage *	% on Target Reads ^§^	1×	5×	10×	20×	30×
Mother (gDNA °)	116X	85.16%	99.63	99.6	99.43	98.52	96.79
Father(gDNA °)	118X	85.87%	99.82	99.65	99.46	99.05	97.60
Amniotic fluid(gDNA °)	70X	82.40%	96.65	92.79	81.76	64.20	60
cff-DNA ^#^	167X	85.34%	99.65	99.50	99.12	97.62	95.16

°: gDNA: genomic DNA; #: cff-DNA: cell-free fetal DNA; *: average sequencing coverage of the whole exome; §: percentage of reads that map in the target regions.

## Data Availability

Not applicable.

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
