# Peer review of "Non-Invasive Detection of a De Novo Frameshift Variant of STAG2 in a Female Fetus: Escape Genes Influence the Manifestation of X-Linked Diseases in Females"

_jcm, 2022, doi:10.3390/jcm11144182_

Round 1
Reviewer 1 Report
REVIEW: Non-invasive detection of a de novo frameshift variant of STAG2 in a female fetus: Escape genes influence the manifestation of X-linked diseases in females. (Note new title!)
This interesting paper presents well documented evidence of the efficacy of prenatal DNA testing from blood samples of pregnant females and is therefore appropriate for this special issue of the journal. It also discusses the influence of escape genes on female phenotype.
There are a few problems that need to be amended.
The first is the paper is not written in the best English. I have made some suggestions for revisions, but perhaps your editors may have other suggestions. Cutting words will strengthen their presentation.
Second, I object to the use of dominant and recessive for X-linked traits. There is a continuum
of expression mediated both by X inactivation, the type of mutation and degree of expression from the Y or inactive X. These authors should avoid interpretating their data as dominant or recessive, or residual, which is used inappropriately, in the paper.
Third, the authors should use the term escape genes, (the proper term), and not escapees or escapers. In addition, they should state somewhere that escape genes have 10-30-% the activity of the gene on the active female X and male X.
Figure 2 is difficult to understand as presented. I still do not see the missing nucleotides.
Defining what is being highlighted would help.
My specific suggestions are as follows:
TITLE: Non-invasive detection of a de novo frameshift variant of STAG2 in a female fetus: Escape genes influence the manifestation of X-linked diseases in females.
Line 35. Omit: on the contrary
Line 36. Missense
Line 39. Omit in determining the X-linked disorders as recessive or dominant.
Line 59. The word are was omitted between which and compatible.
Line 62. Omit semi dominant condition with
Line 221. Use in instead of since
Line 233. Use associate with instead of associate to
Line 236. Omit in particular
Line 245. Omit germinal
Line 247. Use from instead of by
Line 254. Omit sentence starting with Therefore, every possible
Line 257. Omit act as semidominant; Substitute: is another escape gene
Line 258. Omit as and use to be
Line 262. Omit is not an escapee one. Use does not escape X inactivation
Line 264. Omit vary and use varies.
Line 264. Omit A paradigmatic. Use An example
Line 268. Omit no phenomenon of inactivation occurs. Use where X inactivation does not occur.
Line 269. Omit in males compared to females when analyzing expression in the Y homologues. Line 270-271. Omit These data indicate that escapee genes do not totally escape X inactivation
Line 273. Omit residual
Line 275. Omit the senseless sentence. Indeed it is well known that LoF variants can result in partial effect with residual transcriptional activity at least for some transcripts.
Line 277. Omit In agreement and would and semidominant inheritance. Use We expect that expression is related to escape
Line 278. Omit as indicated. Use as shown for DDX3X and intellectual developmental disorders
Author Response
We thank you for providing us the comments on our paper jcm-1814958
REVIEW: Non-invasive detection of a de novo frameshift variant of STAG2 in a female fetus: Escape genes influence the manifestation of X-linked diseases in females. (Note new title!)
This interesting paper presents well documented evidence of the efficacy of prenatal DNA testing from blood samples of pregnant females and is therefore appropriate for this special issue of the journal. It also discusses the influence of escape genes on female phenotype.
There are a few problems that need to be amended.
The first is the paper is not written in the best English. I have made some suggestions for revisions, but perhaps your editors may have other suggestions. Cutting words will strengthen their presentation.
Second, I object to the use of dominant and recessive for X-linked traits. There is a continuum of expression mediated both by X inactivation, the type of mutation and degree of expression from the Y or inactive X. These authors should avoid interpretating their data as dominant or recessive, or residual, which is used inappropriately, in the paper.
Third, the authors should use the term escape genes, (the proper term), and not escapees or escapers. In addition, they should state somewhere that escape genes have 10-30-% the activity of the gene on the active female X and male X.
We added this sentence
These data indicate that escapee genes do not totally escape chromosome inactivation [28] maintaining only a proportion of the activity of the active female X and male X, final expression being mediated by X inactivation, the type of mutation, and the degree of expression in that specific tissue from either the Y or the active X.
Figure 2 is difficult to understand as presented. I still do not see the missing nucleotides. Defining what is being highlighted would help.
Thanks for your suggestion, Figure 2 indicates the ultrasound of the fetus while the two nucleotides deletion is represented in Figure 4. We have improved the resolution and legend of Figure 4
My specific suggestions are as follows:
In accordance with your suggestion, the title is now:
Non-invasive detection of a de novo frameshift variant of STAG2 in a female fetus: Escape genes influence the manifestation of X-linked diseases in females.
Line 35. Omit: on the contrary. done
Line 36. Missense done
Line 39. Omit in determining the X-linked disorders as recessive or dominant. done
Line 59. The word are was omitted between which and compatible. done
Line 62. Omit semi dominant condition with done
Line 221. Use in instead of since done
Line 233. Use associate with instead of associate to done
Line 236. Omit in particular done
Line 245. Omit germinal done
Line 247. Use from instead of by done
Line 254. Omit sentence starting with Therefore, every possible
We modified the sentence “Therefore, every possible relationship between X-inactivation and phenotype remains conflicting [6]. Interestingly, LoF variants in another X-linked gene involved in the co-hesion pathway, SMC1A, which is another escape gene, act as those in STAG2. Both STAG2 and SMC1A are reported as escapee genes [11, 28]. By contrast, missense and LoF variants in the other X-linked gene involved in Cornelia de Lange syndrome, HDAC8, result in severely affected males with either healthy or mildly affected carrier mothers.”
Line 257. Omit act as semidominant; Substitute: is another escape gene done
Line 258. Omit as and use to be done
Line 262. Omit is not an escapee one. Use does not escape X inactivation done
Line 264. Omit vary and use varies. done
Line 264. Omit A paradigmatic. Use An example done
Line 268. Omit no phenomenon of inactivation occurs. Use where X inactivation does not occur. done
Line 269. Omit in males compared to females when analyzing expression in the Y homologues. done
Line 270-271. Omit These data indicate that escapee genes do not totally escape X inactivation
We modified the sentence: “These data indicate that escapee genes do not totally escape chromosome inactivation [28] maintaining only a proportion of the activity of the active female X and male X, final expression being mediated by X inactivation, the type of mutation, and the degree of expression in that specific tissue from either the Y or the active X.
Line 273. Omit residual done
Line 275. Omit the senseless sentence. Indeed it is well known that LoF variants can result in partial effect with residual transcriptional activity at least for some transcripts. done
Line 277. Omit In agreement and would and semidominant inheritance. Use We expect that expression is related to escape done
Line 278. Omit as indicated. Use as shown for DDX3X and intellectual developmental disorders done
Reviewer 2 Report
Dear Authors,
thank you for preparing your paper on molecular diagnosis of a 20-week-old female fetus with a diaphragmatic hernia and other malformations revealed in ultrasound examination. You present a thorough analysis of fetal DNA from two sources – cell free fetal DNA from maternal blood and amniotic fluid, showing that non-invasive prenatal testing may be appropriate for the detection of de novo mutations. I appreciate that your approach takes into consideration the possibility of parental mosaicism and the phenomenon of X chromosome inactivation, making your analysis complete.
I only have a couple of comments you might wish to consider:
1. Lines 93 and 95 – instead of rpm as centrifugal speed units use g force that is independent of rotor size.
2. Figure 4 is very blurred and illegible, so you should either improve its resolution or maybe only describe it in a text.
Author Response
Dear Authors,
thank you for preparing your paper on molecular diagnosis of a 20-week-old female fetus with a diaphragmatic hernia and other malformations revealed in ultrasound examination. You present a thorough analysis of fetal DNA from two sources – cell free fetal DNA from maternal blood and amniotic fluid, showing that non-invasive prenatal testing may be appropriate for the detection of de novo mutations. I appreciate that your approach takes into consideration the possibility of parental mosaicism and the phenomenon of X chromosome inactivation, making your analysis complete.
I only have a couple of comments you might wish to consider:
- Lines 93 and 95 – instead of rpm as centrifugal speed units use g force that is independent of rotor size.
We modified the speed unit using g force
- Figure 4 is very blurred and illegible, so you should either improve its resolution or maybe only describe it in a text.
Thank you for your suggestion, we improved the figure and the legend